# Spatial and temporal expansion of global wildland fire activity in response to climate change

Martín Senande-Rivera [1✉], Damián Insua-Costa [1] & Gonzalo Miguez-Macho [1]

Global warming is expected to alter wildfire potential and fire season severity, but the magnitude and location of change is still unclear. Here, we show that climate largely determines present fire-prone regions and their fire season. We categorize these regions according to the climatic characteristics of their fire season into four classes, within general Boreal, Temperate, Tropical and Arid climate zones. Based on climate model projections, we assess the modification of the fire-prone regions in extent and fire season length at the end of the 21st century. We find that due to global warming, the global area with frequent fire-prone conditions would increase by 29%, mostly in Boreal (+111%) and Temperate (+25%) zones, where there may also be a significant lengthening of the potential fire season. Our estimates of the global expansion of fire-prone areas highlight the large but uneven impact of a warming climate on Earth's environment.

[1] CRETUS Institute, Nonlinear Physics Group, Faculty of Physics, Universidade de Santiago de Compostela, Galicia, Spain.
✉email: martin.senande.rivera@usc.es

Global fire patterns are determined by climate and fuel availability, along with the existence of ignition agents and human factors[1,2]. At the same time, fires also modify the climate through the emission of aerosols and greenhouse gases, and the biosphere by biomass burning, which can lead to deforestation or other modifications of the vegetation structure[3,4]. As a natural process, fire plays a role in some ecosystems, such as being a regulator of biomass in savanna biomes[5]. It is also used as a management tool in pastoral and agricultural areas with regular ignitions from humans[6]. However, fire can be a hazard to the environment, especially during extreme fire events, which have substantial economic, social and ecosystemic impacts[7]. Moreover, landscape fire smoke can be harmful to human health and is an important contributor to global mortality[8].

There is evidence of a greater influence of climate than of human activities on global biomass burning during the Holocene across multiple spatial and temporal scales[9]. Many studies have focused on quantifying the impact of different climatic and human factors controlling global fire activity[10–13]. The variability in global interannual fire response to climate variables has also been previously analysed[14,15]. The general conclusion emerging from these investigations is that climate-related, rather than human factors, are the major controls on global fire activity on a broad spatiotemporal scale, and in particular fuel availability (usually quantified by net primary production, NPP) and precipitation[10,13]. However, on a smaller, regional scale, the drivers of wildfire activity are more varied, and human activity can be the major factor in some areas[16]. Given the strong relationship between fire and climate, climate change resulting from increased greenhouse gas emissions is expected to alter the spatial distribution of fire activity. Some studies point to increases in the severity of the fire season (FS)[17] and the wildfire potential[18], and a gradual shift to a global fire regime dominated by temperature[19], rather than precipitation or human factors, at the end of the 21st century. However, the magnitude and location of change is still debated for many parts of the world[20].

This study aims at (1) demonstrating that through simple climate indicators we can reproduce and explain the present global pattern of fire-prone regions and (2) subsequently use the trends in these indicators to infer future potential changes in the extent of fire activity. We focus on the climate-fire relationship, disregarding interannual variability, ignition elements and human factors (when not related to climate, e.g., some agricultural practices). The underlying hypothesis is that, on broad spatial and decadal scales, there is a high probability of observed fire occurrence wherever a favourable climatic fire setting exists.

## Results

**Present fire-climate classification.** To identify the different regions of the planet with suitable climatic conditions for fire activity, we compare the global distribution of climate indicators based on temperature and precipitation, with satellite-derived GFED4 burned area data[21] (Fig. 1). Starting from four general climates (Tr-tropical, Ar-arid, Te-temperate and Bo-boreal) based on the Köppen–Geiger climate classification main categories[22], we create four fire-prone classes using climate thresholds to define the patterns observed in Fig. 1. Each category is characterised by the element that boosts fire activity during the FS: low precipitation, high temperatures or a combination of both. The classification is made by contrasting the probability distribution of the climatic variables at data points associated with high fire activity vs. points with low fire activity within the main Köppen-Geiger categories (see Threshold Selection in Methods section for a detailed explanation).

The environmental conditions associated with fire occurrence emerge more clearly in this comparison, yielding the different threshold sets in Table 1 that determine the fire-prone months at any location (the selection method is detailed in the Methods section). We define those years with at least 1-month meeting the thresholds, as fire-prone years (FPY). Depending on the number of FPY at each location, the categories of Table 1 are sub-divided into recurrent (r), occasional (o) and infrequent (i) (see Methods). The average number of fire-prone months during the FPY is defined as the potential FS length (PFSL), i.e., the season with climatic characteristics prone to fire activity.

Figure 2a depicts the global map of the burned areas classified according to the selected thresholds (Table 1). Savanna fires are responsible for the largest proportion of burned area on the global scale[21]. The FS in these areas is longer than in other climates (see Supplementary Fig. 1) and, despite savanna fires being also dependent on ignition patterns and human policies and practices, the FS is tied to a pronounced seasonal cycle of precipitation[23–25], with fire occurring mainly during the dry part of the cycle. Because of this, the Tropical - dry season fire class (Tr-ds) coincides with the distribution of the tropical savanna climate. In Fig. 2, boreal fires are represented as hot season fires (Bo-hs) due to the large positive temperature anomaly existing in those locations during the FS (Fig. 1c). In fact, temperature variations explain much of the variability in boreal burned area[26,27]. Temperate fires are classified as dry and hot season (Te-dhs) because they affect regions where the dry season coincides with the warm season (Fig. 1b, c). Here, high temperatures and precipitation seasonality determine fire activity and inter-annual burned area variability, e.g., in Western North America[28–31] and Southern Europe[32,33]. Fire activity in arid regions occurs during warm months, but the relation with precipitation is more complex. The FS is associated with a hot season in cooler (MAT < 18.5 °C) midlatitude arid areas where no clear wet period is observed, e.g., the Western US and Central Asia (Supplementary Figs. 10 and 11), but closer to the tropics where it is warm year-round, it can be also determined by the existence of a marked annual wet and dry season cycle, with fires occurring sometime during the dry season. In the warmest arid regions (MAT > 27.5 °C), the FS starts right at the beginning of the dry season (e.g., the Sahel, Supplementary Fig. 12) while where MATs are more moderate, between 18.5 and 27.5 °C, it takes longer to develop (e.g., Central Australia and the Kalahari desert, Supplementary Figs. 12 and 13). Due to the dependency between fires and the existence of fuel in arid climates, we named this class Arid fuel limited (Ar-fl). A more in-depth discussion about the definition of this fire-climate class can be found in the section entitled Threshold selection for each climate of the Supplementary Information.

In Fig. 2b, we classify every spatial point and not only those with burned area observations as in Fig. 2a. The four groups in Fig. 2a show observed fire locations that share some specific climate conditions, and the classification in Fig. 2b shows the world areas where these conditions occur for at least one month. The correspondence between these two maps is quantified in Supplementary Fig. 20b, with more than 70% of the land area well classified as either fire-prone (BA > 0 ha) or fireless (BA = 0 ha). This reveals a two-way relation between fires and climate: fires take place under specific climatic conditions, and most places with these climatic conditions are indeed fire-prone, which supports our earlier hypothesis. Fire activity is controlled by weather, resources to burn and ignitions, as represented through the fire regime triangle[12,20]. On broad temporal scales and large spatial scales, temperature and precipitation have an important impact on fire because these climate variables influence vegetation type and the abundance, composition, moisture content, and structure of fuels[34]. Although ignitions may be driving fires to a greater extent than temperature or precipitation at specific locations or events[35], they do not seem to limit fire activity at coarse spatial and temporal resolutions, implying that where fuels

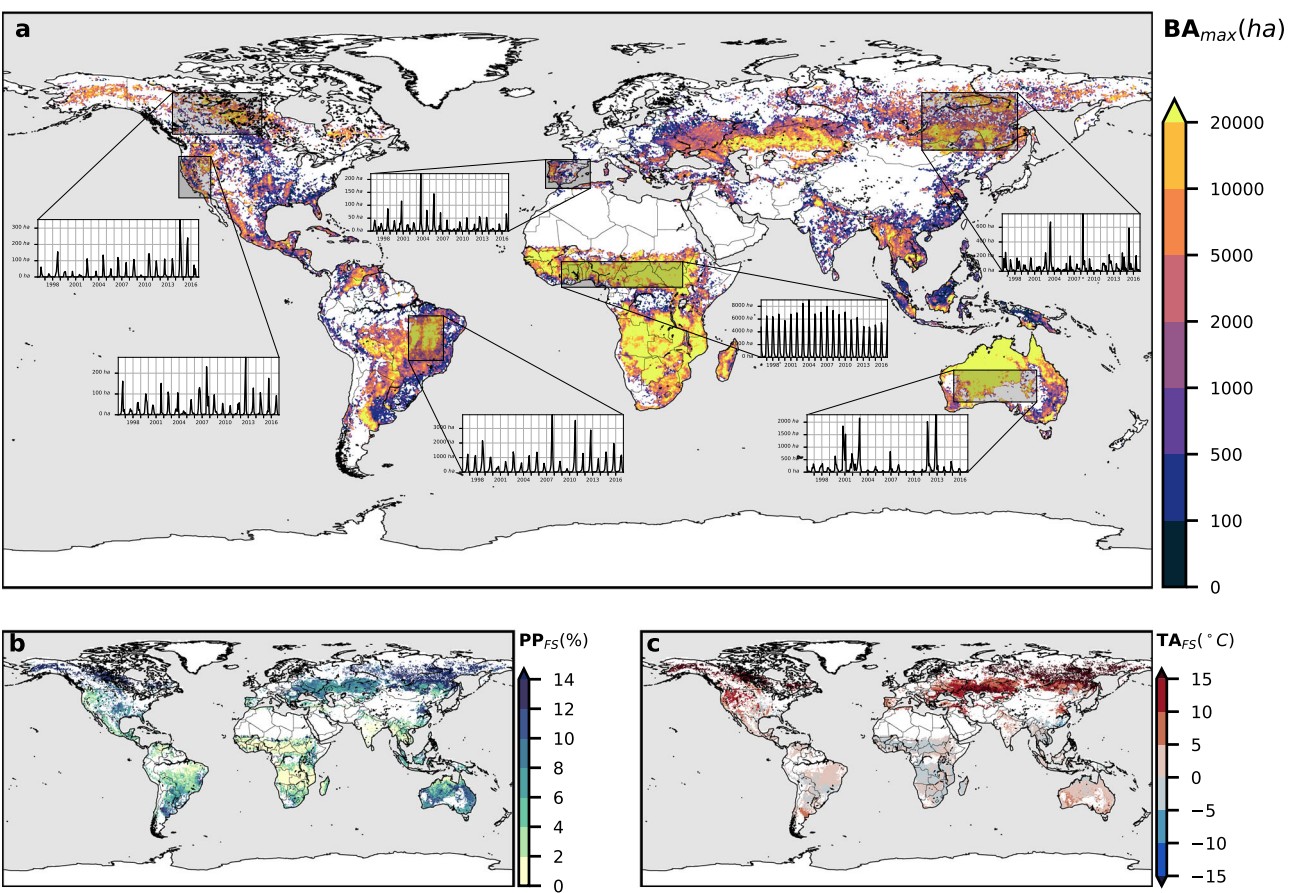

**Fig. 1 Burned area observations and climate drivers. a** 1996–2016 maximum annual burned area ($BA_{max}$) and monthly burned area time series for selected regions. **b** Average monthly precipitation percentage from the annual total for the fire season ($PP_{FS}$). **c** Average monthly temperature anomaly from the annual mean for the fire season ($TA_{FS}$).

**Table 1 Fire classification defining criteria.**

| | |
|---|---|
| **Tr** | **MAP ≥ 10 · P$_{threshold}$ & Tcold ≥ 18 °C** |
| Tr - ds | Pa ≥ 220 mm and P$_{min}$ ≤ 6 mm and P$_m$ ≤ 90 mm |
| **Ar** | **MAP < 10 · P$_{threshold}$** |
| Ar - fl | Pa ≥ 220 mm & Tm ≥ 19.5 °C & Pm ≤ 60 mm |
| | if MAT < 18.5 °C, all months analysed |
| | if 18.5 °C ≤ MAT < 27.5 °C, months with |
| | 7 ≤ dm$_{Pmax}$ ≤ 10 analysed |
| | if MAT ≥ 27.5 °C, months with 2 ≤ dm$_{Pmax}$ ≤ 4 analysed |
| **Te** | **MAP ≥ 10 · P$_{threshold}$ and Tcold < 18 °C & MAT ≥ 2 °C** |
| Te - dhs | Pa ≥ 220 mm and Pmin ≤ 13 mm and Tm ≥ 12 °C and Pm ≤ 42 mm |
| **Bo** | **MAP ≥ 10 · P$_{threshold}$ and MAT < 2 °C** |
| Bo - hs | Pa ≥ 220 mm and Tmax > 15 °C and Tm ≥ 7 °C and Pm ≤ 67 mm |

Fire-climate classes (Tr-ds, Ar-fl, Te-dhs and Bo-hs) must meet general Köppen–Geiger climate criteria (Tr, Ar, Te and Bo, in bold) and the specific fire-climate thresholds. Fire-climate classes are subdivided into three sub-classes according to the number of years classified per decade: r-recurrent (FPY > 7 years/decade), o-occasional (7 > FPY > 3 years/decade) and i-infrequent (3 > FPY > 0 years/decade). All variables are defined in Table 2.

are sufficient and atmospheric conditions are conducive to combustion, the potential for ignition exists, either by lightning or human causes[13,20]. For all these reasons, we can identify specific climates that are prone to fires.

The areas classified as fire-prone in Fig. 2b comprise 99.26% of the observed global mean annual burned area in Supplementary Fig. 2. This percentage is above 85% for all four general climates (Supplementary Fig. 20). The percentage of land area with non-zero burned area data classified as fire-prone is 91.22%. Considering for each location only the obtained FPY, the percentage of the observed burned area classified is 90.36%. Furthermore, the PFS obtained in the fire-climate classification (Fig. 3b) also correlates well with the timing of observed fire incidence, as globally 87.91% of the observed mean burned area occurs during the identified months of PFS at classified fire-prone locations.

Unclassified regions (in grey in Fig. 2a) correspond for the most part to those with the least burned area or those where agricultural practices modify the climatic seasonality of fires. In addition, as the classification is conceived from a climatic point of view, locations with fire activity associated with specific meteorological conditions that are not appreciable at the monthly temporal resolution, are probably not well identified. For example, a week of extremely high temperatures could be almost unnoticeable in the monthly mean temperature, but not in fire activity. Similarly, months with the same total precipitation may have different fire activity if the precipitation falls concentrated in a few days or is distributed throughout the month. Furthermore, the short temporal sampling period of the burned area data could also be influencing our results. Locations with long fire cycles may not be well represented in the data.

**Future fire-climate classification.** A future fire-climate classification map is derived by applying the thresholds obtained in the

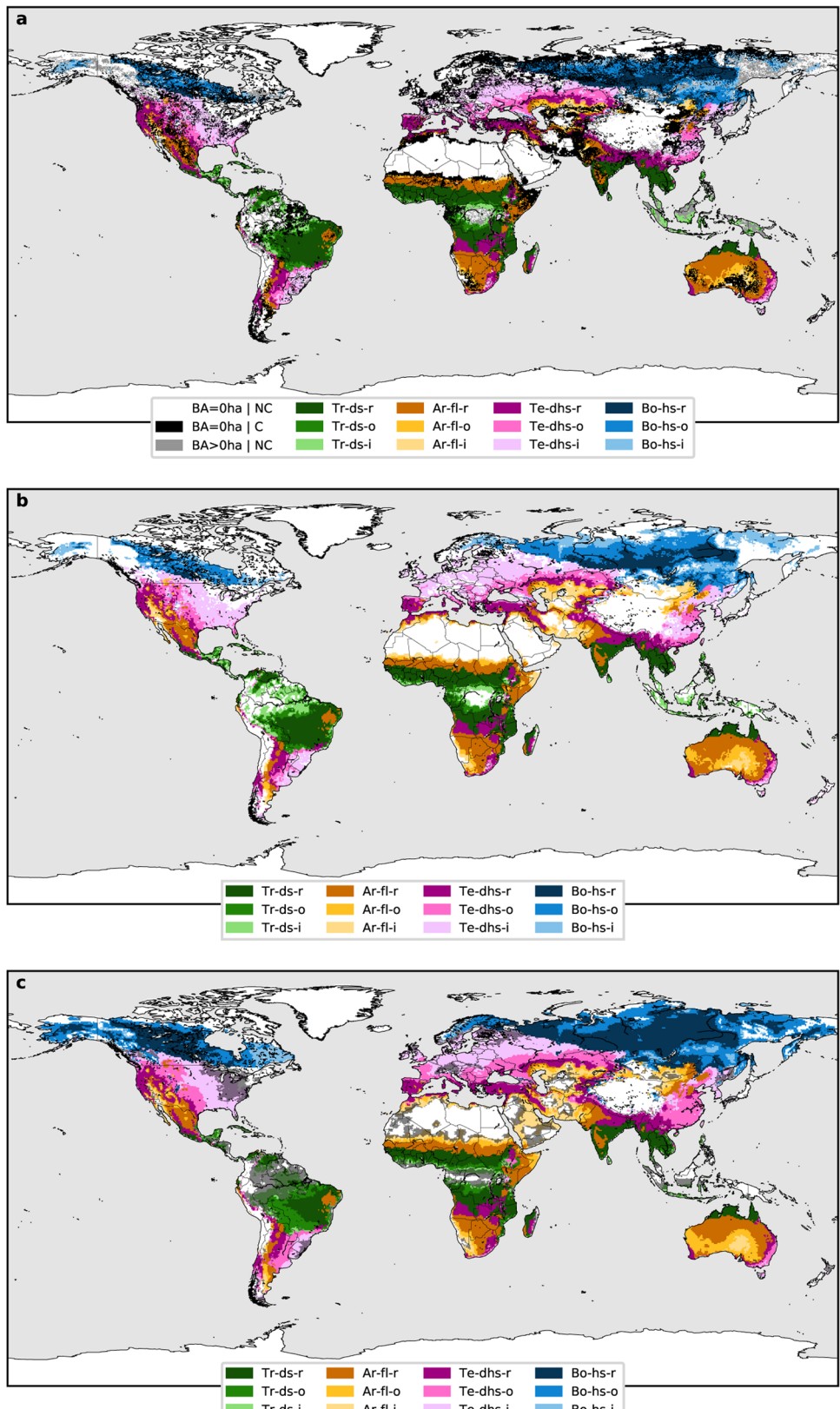

**Fig. 2 Fire-prone region classification. a** With observed burned area data as a reference: not classified (NC, white) and misclassified (C, black) areas with $BA_{max} = 0$ ha, unclassified (NC, grey) and classified (Tr-ds, Ar-fl, Te-dhs and Bo-hs) areas with $BA_{max} > 0$ ha. Each class is subdivided into three subcategories depending on the recurrence of the fire-prone conditions: recurrent (r), occasional (o) and infrequent (i). **b** Present (1996–2016) fire-prone climatic regions. **c** Future (2070–2099) fire-prone climatic regions with shaded grey representing a <75% confidence percentage, estimated as the percentage of CMIP5 Global Circulation Models (GCMs) agreeing on the result.

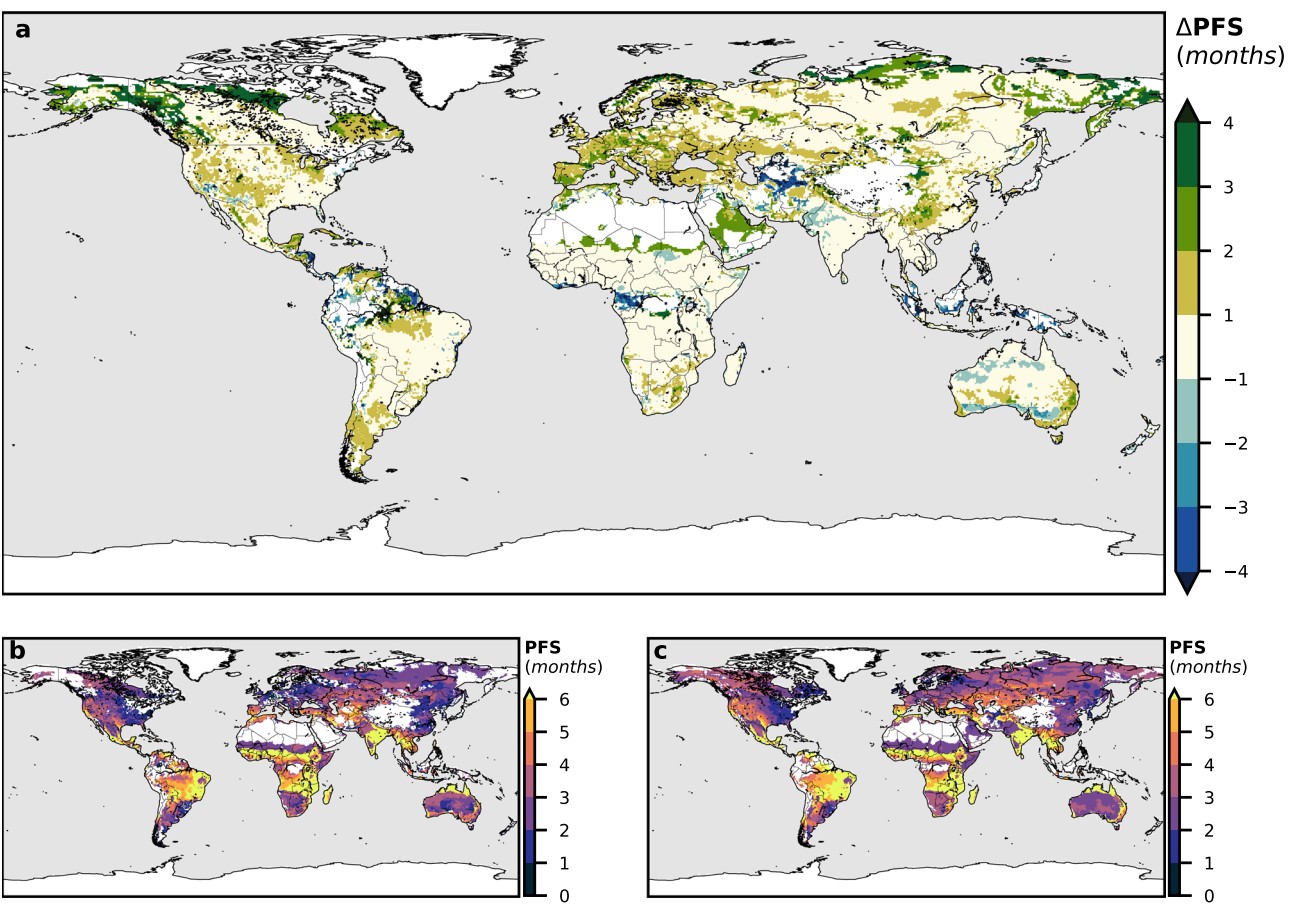

**Fig. 3 Potential fire season. a** Future minus present potential fire season length (PFSL) difference in months (ΔPFSL). **b** Present potential fire season. **c** Future potential fire season.

present fire-climate classification to future climatology variables from multiple coupled model intercomparison project phase 5 (CMIP5) global circulation model (GCM) outputs, considering the RCP8.5 scenario (the worst-case climate change scenario of the CMIP5). Two contrasting approaches can be taken for analysing future fire activity, one that considers quick vegetation adaptation to the new climatic conditions, and another that does not. These two approaches clearly diverge in the boreal regions, where the biome (mainly taiga) is strongly conditioned by the low temperatures and where future temperature changes at the end of the 21st century will have a greater amplitude. It is expected that the boreal forest of these areas will not be immediately replaced by a temperate mixed forest where the average annual temperature exceeds the range of values typical of the taiga biome. Terrestrial vegetation compositional and structural change could occur during the 21st century where vegetation disturbance is accelerated or amplified by human activity, but equilibrium states may not be reached until the 22nd century or beyond[36].

Based on the assumption that during the future period (2070–2099) the vegetation will not be fully adapted to the new climatic conditions, and since the present Köppen–Geiger climate classification (on which we base our Tr, Ar, Te and Bo categories) closely corresponds to the different existent biomes[22], we analyse only the projected changes in the specific fire-climate classification variables, maintaining the general division of Tropical, Arid, Temperate and Boreal regions as is in present climate conditions. The future fire-climate classification is shown in Fig. 2c.

We note that we determine future fire activity from relationships of the latter with the present climate; however, these relationships might not be stationary. Our approach does not

contemplate possible future changes in precipitation frequency if they are not noticeable in monthly precipitation amounts. Areas with the rising incidence of extreme precipitation events due to global warming[37] may experience an increase in monthly precipitation but a decrease in rainy days, which may lead us to consider the conditions there less favourable for fire activity than they actually will be.

**Future changes in global fire activity**. Modelled future fire-prone regions experience significant variations with respect to the present (Fig. 2b, c). Due to global warming, the Bo-hs fire class pertaining to boreal forests would spread over a larger area, reaching most of Northern Scandinavia and undergoing a southward and northward expansion in Canada, Alaska and Russia. This category may experience a percentual expansion of 47.0% according to our results. This expansion is more accentuated for the combination of the highest recurrence subcategories Bo-hs-r and Bo-hs-o, reaching a value of 111.5%.

The conjunction of Te-dhs-r and Te-dhs-o fire classes of midlatitudes also undergoes a considerable expansion of 24.5% in the area (Fig. 2b, c). The most remarkable changes are expected in Southern China and Southern Europe. A large part of Europe transitions from an infrequent fire category to a more frequent fire category with Csa and Csb Mediterranean climates[38].

The Tr-ds fire classes with frequent fire-prone conditions in the Tropics presents fewer spatial changes (Fig. 2b, c), with a spatial contraction of 6.3%. The most important differences are found in South America. Some of the climate model results considered here indicate also that some parts of the Eastern Amazon

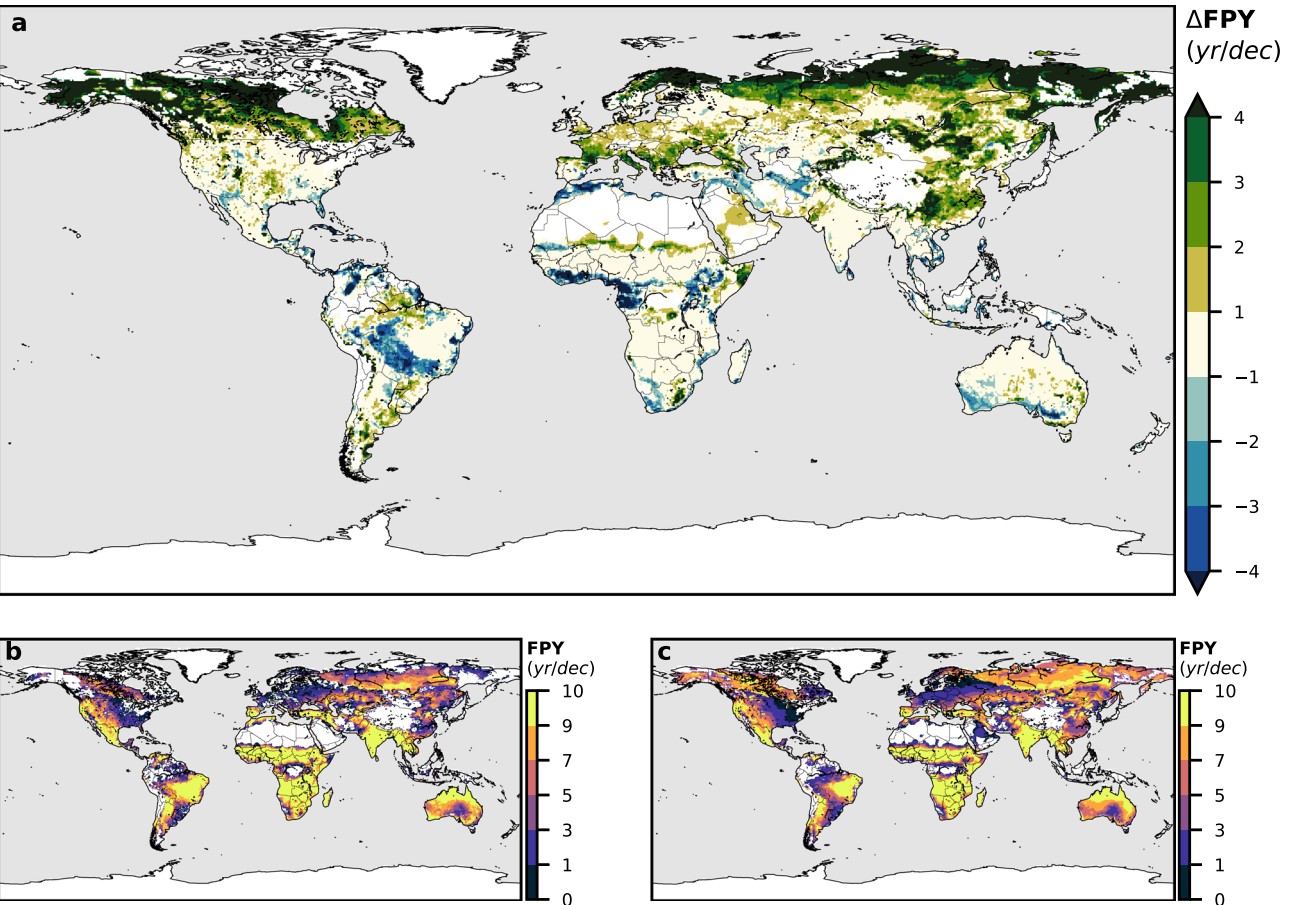

**Fig. 4 Fire-prone years. a** Future minus a present number of years with at least one month classified as fire-prone per decade (ΔFPY). **b** Present fire-prone years per decade. **c** Future fire-prone years per decade.

rainforest will move from a non-fire class to Tr-ds fire class, as other studies have suggested[39].

The Arid fire-prone classes Ar-fl-r and Ar-fl-o would increase its area by 5.0%. Projected changes in the extent of this class are very sensitive to changes in annual precipitation, conducive to vegetation and fuel reduction or increment, thus there is significant uncertainty in the proximity of desert regions (Fig. 2c).

Clearer conclusions can be drawn from the FPY and PFSL calculation (Figs. 3 and 4). The number of months meeting the set of conditions in Table 1 yields the estimated PFSL (Fig. 3b), and the number of years with at least 1-month meeting the thresholds, the FPY. In the boreal regions, we obtain a general lengthening of the PFS. The PFS of these areas is conditioned by temperature, so the amplified warming of Artic zones[40] is expected to make the FS longer. Notwithstanding, in certain parts of Eastern Asia, the intense warming is counterbalanced by an increase of the precipitation in certain warm months (see Supplementary Figs. 21 and 22), leading to a slight shortening of our estimated PFS. There is evidence, however, that temperature increases may lead to drier fuels in the future despite the precipitation increase, thus augmenting fire risk, as some investigations have shown for Canada[41]. Our results agree in general with several other studies that have previously pointed towards an increase of the FSL in boreal areas[1,17,42], even when some suggest a more pronounced lengthening in more northerly latitudes[1,17]. In terms of the frequency of years with fire-prone conditions, the conclusions are even clearer. A general increase of the FPY is observed, especially for northerly latitudes, where the differences reach values of more than +4 years per decade

(Fig. 4a). This possible increase in fire activity in boreal areas may result in significant peatland combustion and a release of the large quantities of soil carbon that they store into the atmosphere[43]. These greenhouse gas emissions may create a positive feedback loop, leading to a further increase in temperature, which in turn will enhance boreal wildfire incidence and more peatland burning.

The Te-dhs fire class, corresponding to temperate climates, would also experience a general lengthening of the PFS (Fig. 3). A future precipitation decline may be especially significant in Southern Europe (Supplementary Fig. 21), associated with an increased anticyclonic circulation yielding more stable conditions[44], while the temperature rise would be quite homogeneous among all Te-dhs fire-climate class areas. The FS drought intensification around the Mediterranean, together with the general warming (Supplementary Fig. 21), would lead to a lengthening of the PFS of around 2 months (Fig. 3a), but summer months could also experience this precipitation decline (Supplementary Fig. 22), meaning that the FS would be more severe. The Western US, which has already experienced over the last decades the lengthening of the FS[45] and the increase of large fires[46] and extreme wildfire weather[47,48] due to climate change, may also experience an FS lengthening by the end of the 21st century. Some authors[18,48–50] have studied projected fire future changes from other points of view (occurrence of very large fires, wildfire potential, etc.), finding also a general increase of fire severity by the end of the century in some of these Te-dhs fire regions. The interannual recurrence of fire-prone conditions will significantly increase in countries like France, Italy or Eastern China (Fig. 4a).

The PFSL of the Tropical Tr-ds fire-climate class presents slight differences between present and future values (Fig. 3). Some areas of the Northern African savanna may experience a shortening of the PFS, while Southern Africa shows a lengthening. A dipole pattern of wetting in tropical Eastern Africa and drying in Southern Africa[51] could be the reason for these future changes. There is a contrasting influence of ENSO in present African fire patterns[52], which suggests that the future pattern of precipitation variations in Central Africa may be associated with ENSO future changes under climate change conditions[53]. Although the quantification of ENSO changes in a warmer climate is still an issue that continues to be investigated, an expansion and strengthening of ENSO teleconnections is confirmed by some authors[53–55]. The general increase in precipitation along all seasons in western equatorial Africa would lead to a significant decrease in the recurrence of interannual fire-prone conditions (Fig. 4a).

Our results show that fire-prone areas in Temperate and especially Boreal climates are projected to undergo the most significant expansion and lengthening of the potential FS at the end of the XXI century driven by rising temperatures. In the Tropics, little change is expected in these respects. Notwithstanding, global warming is likely to make fire risk more severe mostly everywhere, and in particular in some regions such as Mediterranean Europe and the Eastern Amazon, where an important decrease in precipitation is also predicted during the PFS. More favourable fire conditions will potentially increment fire activity and burned areas in many places. In others, especially in the Tropics, increasing suppression efforts and a cease to agricultural and pastoral practices like vegetation clearing by fire, replaced by more intensive farming, could counteract the impact of a warmer climate. A reduction of these human-caused fires in the Tropics could bring global burned area down[2], despite rising trends elsewhere, given the vast contribution of Tropical fires to the burned areas at the global scale (Fig. 1).

## Methods

**Burned area data**. To study how climate influences fires at the global scale we use the Global Fire Emissions Database (GFED)[56]. The fourth generation of the GFED burned area data set (GFED4) provides global monthly burned area at the 0.25° spatial resolution from mid-1995 to 2016[21]. This dataset is obtained from the Collection 5.1 MODIS direct broadcast (DB) burned area product (MCD64A1), now generated globally using the MODIS DB burned-area mapping algorithm[57].

**Climate data**. Climate datasets are acquired from WFDE5[58], WATCH Forcing Data methodology applied to ERA5[59], the fifth generation ECMWF reanalysis for the global climate. This is a meteorological forcing dataset for land surface and hydrological models. The dataset was derived applying sequential elevation and monthly bias correction methods[60] to 0.5° aggregated ERA5 reanalysis products[61]. The monthly observational datasets used for bias correction are the Climate Research Unit gridded station observations CRU TS4.03[62] and the Global Precipitation Climatology Centre gridded station precipitation observations GPCCv2018[63] for rainfall and snowfall rates. From this climate dataset, we calculated the annual and monthly temperature and total precipitation (rainfall and snowfall) climatology for the same 1996–2016 period. We downscale these variables to a 0.25° grid using a bilinear interpolation.

**GCM data**. For analysing future variations in fire activity, we use the Coupled Model Intercomparison Project Phase 5 CMIP5[64] historical and future projections from 22 climate models. The GCMs are ACCESS1-0, ACCESS1-3, bcc-csm1-1, bcc-csm1-1-m, BNU-ESM, CMCC-CMS, CNRM-CM5, GFDL-CM3, GFDL-ESM2G, GFDL-ESM2M, GISS-E2-H, GISS-E2-H-CC, GISS-E2-R, GISS-E2-R-CC, HadGEM2-CC, inmcm4, IPSL-CM5A-LR, IPSL-CM5A-MR, IPSL-CM5B-LR, MPI-ESM-LR, MPI-ESM-MR and NorESM1-M. We use the RCP8.5[65] greenhouse gas concentration model outputs for the future period. The Representative Concentration Pathway RCP8.5 assumes a scenario in which greenhouse gas emissions continue to increase throughout the 21st century, which leads to a radiative forcing of 8.5 W m$^{-2}$ by 2100[66]. For each model, we only consider the main ensemble member (r1i1p1). We analyse two periods, present (1996–2016) and future (2070–2099). The present period is obtained by concatenating precipitation and temperature historical data (1996–2006) and projected data until 2016 (2006–2016). Future climatology values are obtained with the anomaly method[67]: for each one of the 22 CMIP5 global models, we calculate temperature anomalies and precipitation ratios between 1996–2016 and 2070–2099 and interpolate them from their native model resolution to 0.25° using bilinear interpolation. Finally, future values of the variables of interest were derived from present values by adding the temperature anomalies to present temperatures and by multiplying the present precipitation by the precipitation ratios.

**Threshold selection**. We examine the statistical distribution of different climatic variables at fire-impacted and fireless pixels within the Köppen–Geiger main categories, to find the conditions that are suitable for fire activity. The objective is to obtain sets of climatic thresholds that univocally identify the different classes of burned areas in actual fire occurrence data, both in extent and duration of the FS. Precisely, for each grid cell we obtain the mean value of each annual variable (Pa, Ta, Pmin and Tmax) for the years with an annual burned area (BA) greater or equal than 100 ha (we will refer to all these data values as fire "points"), and the mean value for the years with BA < 100 ha (which we will refer to as non-fire "points"). Once the data values are separated into the fire and non-fire groups, we compare for each main Köppen–Geiger category (Supplementary Fig. 3) the statistical density function for BA ≥ 100 ha values against BA < 100 ha values.

We next identify the FS months for each grid cell using mean monthly BA computed over the years with BA > 100 ha. We start from the month with the highest BA and proceed with lower BA months until at least 80% of the total mean BA at the location is included. The FS months do not need to be contiguous.

For the monthly variables (Pm and Tm) we make two different comparisons. We first contrast their average magnitude during each FS month for years with BA ≥ 100 ha (fire "points") against average values in months out of the FS for all years (non-fire "points"). At grid cells with no fire incidence (BA < 100 ha in all years), there is no FS, and means for each and every month, computed over all years, are added to the non-fire "points". With this analysis we assess the seasonal variability of the monthly variables, determining the values that characterise the FS. We then make a second comparison using the monthly variables aiming at analysing the interannual variability of the FS. Here, we compare average Pm and Tm values for the FS months of the years with BA ≥ 100 ha (fire "points"), against averages for FS months in years with BA < 100 ha (non-fire "points"). By doing so, we check whether the FS months present precipitation or temperature differences between years with fire activity and years with no fire activity. These interannual differences are only detectable if variability is an intrinsic characteristic of the climate. Supplementary Fig. 4 shows a scheme of the three comparisons of statistical distributions: the annual variables, the seasonal analysis and the interannual variability analysis

We note that we exclude from the statistical distributions data from fire-prone regions with a percentage of the mean annual burned area corresponding to cropland land cover of more than 90% (Fig. S1b), due to the likely relation of fires with agricultural practices in these grid cells[67–69].

The thresholds are selected automatically for each distribution at the value that maximises the area enclosed between the density function of the fire points and the density function of the non-fire points (coloured minus grey areas in Supplementary Figs. 5, 6, 8, 9, 16–19). The only prior consideration is that the thresholds work as an upper limit for Pmin and Pm variables (looking for seasonal dry conditions), and as a lower limit for the other variables (looking for warm conditions and fuel availability). Distributions with a big area difference evidence that the considered variable is a good indicator of fire incidence since the values associated with fire activity are clearly distinct from the values that are not. In an ideal scenario where the data points associated with fire activity, or fire points, were perfectly separated from the remaining points, this area would be equal to 1. In most cases, however, it is the simultaneous meeting of the conditions for two variables that determines fire risk; thus, there can be a sizeable amount of non-fire data points satisfying either of them. The objective is then that fire points, and especially those representing locations and months with the more burned area, fulfil all the conditions simultaneously in the highest percentage possible, while non-fire points that do so are a minority.

Once we have a threshold for each distribution, we select the variables and thresholds for the classification according to the area difference and classified fire-point percentage criteria. The final threshold is chosen as a rounded value inside the obtained uncertainty range. More information about the threshold selection method for each general climate can be found in the Threshold selection for each climate section of the Supplementary Information.

Tables 1 and 2 show all defining criteria for the four classes, including the Köppen-Geiger-based general classification of climates zones and the specific fire classification thresholds. One grid cell is classified as fire-prone if at least one month in the time series meets all the conditions. For each cell, we define FPY as a year with at least 1-month meeting all the conditions in Table 1. The months that meet the thresholds during a particular FPY conform to the PFS. The categories of Table 1 are sub-divided into: recurrent (r), occasional (o) and infrequent (i) depending on the number of FPY per decade: recurrent (FPY > 7 years/decade), occasional (3 < FPY < 7 years/decade) and infrequent (0 < FPY < 3 years/decade).

**Table 2 Variable definitions.**

| Variable | Units | Definition | Temporal dimension |
|---|---|---|---|
| MAT | °C | Mean annual 2 m air temperature | – |
| Ta | °C | Annual 2 m mean air temperature | Years |
| Tcold | °C | Mean 2 m air temperature of the coldest month | – |
| Tmax | °C | Maximum monthly 2 m mean air temperature of the year | Years |
| Tm | °C | Monthly 2 m mean air temperature | Months |
| MAP | mm | Mean annual precipitation | – |
| Pa | mm | Annual precipitation | Years |
| Pmin | mm | Minimum monthly precipitation of the year | Years |
| Pm | mm | Monthly precipitation | Months |
| $P_{threshold}$ | mm | $2 \times MAT$ if $P_{winter} > 70\%$<br>$2 \times MAT + 28$ if $P_{winter} < 30\%$<br>$2 \times MAT + 14$ otherwise | – |
| $P_{winter}$ | % | Percentage of MAP that falls during the colder six-month period between April-September and October-March[34] | – |
| $dm_{Pmax}$ | – | Month distance to the wettest month of the year, ranging from 0 to 11 | Months |

## Data availability

The data generated in this study have been deposited in the Fire-Climate classification database (https://doi.org/10.7910/DVN/J31ZBD).

## Code availability

The codes used to obtain the classification thresholds and to classify the climate data have been deposited in the Fire-Climate classification database (https://doi.org/10.7910/DVN/J31ZBD).

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

## Acknowledgements

Funding comes from the Spanish Ministerio de Economia y Competitividad OPERMO (CGL2017-89859-R) and the CRETUS strategic partnership (AGRUP2015/02). All these programs are co-funded by the European Union ERDF. M.S.R. acknowledges Xunta de Galicia for a predoctoral grant (Programa de axudas á etapa predoutoral 2019, ED481A-2019/112). D.I.C. was awarded a pre-doctoral FPI (PRE2018-084425) grant from the Spanish Ministry of Science, Innovation and Universities. Computation took place at CESGA (Centro de Supercomputacion de Galicia), Santiago de Compostela, Galicia and Spain.

## Author contributions

M.S.R. designed the study and drafted the paper. D.I.C. and G.M.M. provided critical feedback and contributed to writing the paper.

## Competing interests

The authors declare no competing interests.
