## [Peer Review File · Nature Communications]

REVIEWER COMMENTS

Reviewer #1 (Remarks to the Author):

The authors present an analysis of the climatic conditions that describe fire season lengths throughout the world and they leverage that analysis to explore how climate change may impact the length of future fire seasons and expansions of fire-prone areas. I feel that the paper is well-written and most of the analysis is well-described but there are some potential aspect of this analysis that I feel need further examination.

General Comments:

While in general, I found the analysis interesting but the use of a thresholded mean annual burned area to identify fire prone areas is very problematic because it ignores the basic fire ecology of many of these systems. Given the short time series of satellite burned area data, many areas with long fire return intervals would only likely see fire once in the entire record. That means if you average the annual burned area, many places would be characterized as having a small mean burned area but they are indeed very fire prone. Only places that burn frequently, like African savannas would appear to have a high mean annual burned area. I am concerned that this method failed to classify most of Alaska and Eastern Siberia, the Entire Eastern United States, Indonesia, and Southeastern Australia as fire prone. Some of these areas have infrequent but high intensity fires that heavily impact ecosystems and people. This suggests a methodology flaw that must be corrected, especially since the main conclusions about how the fire prone areas have increased shows an increase in Boreal areas such as Alaska where fire is already common and widespread. It is likely that the using a maximum annual burned area would give you a different result.

Further, the use of mean annual burned area heavily biased the claim of 94.7% of fire prone areas being classified because of the overwhelming amount of burned area in regions like Sub-tropical Africa. Simply classifying areas as burned or unburned, rather than as a percentage of their mean annual burned area, would likely vastly reduce that classification accuracy and would highlight this deficiency.

In the Supplement, the authors detail the method used discriminating the points with and without fire activity and they state that the thresholds were selected 'automatically' at the point that discriminates the two distributions. However, they failed to detail HOW this process works. As such, I do not believe that someone else could reproduce these results from this explanation. Additional clarification is required to ensure results are reproducible.

Finally, the definition of fire season length very likely won't work for places that have a bi-modal fire season. In many places

Specific Comments:

The use of A,B,C and D in the manuscript is difficult to follow and I found I needed to flip back and forth a lot to interpret parts of the manuscript. I recommend using the first two letters of the climate type (Tr for Tropical, Te for Temperate, Ar for Arid and Bo for Boreal or something similar. It should help increase readability throughout the manuscript and the supplemental information.

In the Supplementary Material, it would be more readable if the authors provided complete figure captions rather than stating "Same as Fig. S3".

Reviewer #2 (Remarks to the Author):

This manuscript explores the temporal and spatial changes of global wildfire activity as a result of climate change. This is an interesting paper.

One area that needs attention is the distinction between weather and climate. Fire activity in many places depends on the day-to-day weather and is often determined by extreme conditions on a small number of days. Monthly variables may not always be good indicators of fire activity in some regions. For example, the monthly precipitation amount is not as important as frequency of precipitation. I do not require the authors to change the temporal scale of your analysis but rather acknowledge that the temporal scale may be too coarse in some regions where the bulk of the fire activity may occur over a very short time period associated with extreme conditions.

Did you consider looking at vapor pressure deficit (VPD) that has been found to be related to fire activity? VPD drives the dead fuel drying process.

Inferring future change in wildland fire activity using present relationships may not work due to the likelihood that the relationships found are not stationary. This needs to be acknowledged. Additionally, there are lots of cells with no area burned due to a very short temporal sampling period, so this data is censored. For cells with no area burned, we do not know if the fire cycle is 30 years or 10,000 years. This limitation needs mention.

Editorial

Title should use wildland fire or vegetation fire instead of just fire.

Introduction

Might mention the role of wildland fire smoke on human health (Johnston et al. 2012).

L57-58 these are fireless months during the fire season?

L74 Savanna fires are also dependent on ignitions patterns (temporal and spatial)

As well as local/national policies and practices.

L104 supports rather than confirms

L133 RCP8.5 – only RCP used, so should acknowledge that this is on the extreme end of the CMIP5 scenarios.

Johnston FH, Henderson SB, Chen Y, Randerson JT, Marlier M, Defries RS, Kinney P, Bowman DM, Brauer M (2012) Estimated global mortality attributable to smoke from landscape fires. *Environmental Health Perspectives* 120(5), 695–7

Reviewer #3 (Remarks to the Author):

Main paper

The manuscript “Spatial and temporal expansion of global fire activity in response to climate change” proposes a classification of potential fire season (PSL) based on temperature and precipitation and uses this classification to project future PSL. In general, this seems like an interesting and potentially useful approach and the results seem both interesting and relevant. However, I believe this paper needs restructuring since the most interesting aspects of the paper are currently buried in the supplemental, which describes in detail the classification of the PSL for each region. I understand Nature CC has word restrictions which may have necessitated moving many of the details to supplemental; however, the purpose of supplemental is not to supplant the main text due to word limits. Indeed, the methods section (lines 381-382) refers the reader to the supplemental for all details of threshold selection, and Tables 1 and 2 lack context in the main paper with a better description in the main text.

Other general comments:

The manuscript considers precipitation and temperature thresholds to calculate potential fire season. Did the authors also consider atmospheric humidity or surface wind speeds in their analysis, which are well-known to correlate with fire danger? (see Canadian Fire Weather Index system for example).

It is not clear what the spatial units are for the modeling of probability density functions for climatic variables (and assessment of thresholds). It seems to be done at each of the four main regions (A, B, C, and D), which is rather broad spatially. However, the supplemental includes figures S5-S8 that show the associations for smaller spatial units. How robust are the results to choice of these spatial units given that fire activity and climate are both heterogenous even within a single region (eg. each A, B, C or D category). Did the authors consider using smaller spatial units, while understanding the spatial units should be large enough to encompass a well-defined fire season.

The climate projections presented here assume that monthly precipitation changes will dictate future PFSs within each category. However, much of the literature now points towards changes in both intensity and frequency of future precipitation events (eg. Myhre et al. 2019). How would changes in precipitation frequency change the results reported here? The authors should address this, at least in the discussion.

Myhre, G., Alterskjær, K., Stjern, C.W. et al. Frequency of extreme precipitation increases extensively with event rareness under global warming. *Sci Rep* 9, 16063 (2019). <https://doi.org/10.1038/s41598-019-52277-4>

Additional minor comments:

Lines 18: What impacts do wildfires have? It wouldn't hurt to elaborate here.

Lines 20-21: "biosphere by biomass burning" seems quite vague – what changes to the biosphere are occurring?

Line 24: "Fire can be a hazard..." reads better as "However, fire can be a hazard..."

Lines 27-28 and lines 31-34: I would say these statements are not entirely true since the dominant controls on wildfire are very regionally dependent; in many regions of Asia and Africa fire depends predominantly on human activity since many of the ignitions are human-caused. For example see:

Forkel, M., Dorigo, W., Lasslop, G., Chuvieco, E., Hantson, S., Heil, A., ... & Harrison, S. P. (2019). Recent global and regional trends in burned area and their compensating environmental controls. *Environmental Research Communications*, 1(5), 051005.

Lines 45-47: Fire occurrence is often ignition- or fuel-limited despite favourable climatic conditions; I would suggest the authors emphasize that they are talking about over a sufficiently long time period (ie. Decades).

Fig. 1b. I would suggest not using a diverging colormap for this panel since the quantities do not diverge around zero. Also, with respect to this plot, why not show precipitation anomalies instead of percentage difference?

Lines 68-72 – why is it important to distinguish between FS and PFS here and what are the implications? This text seems like an unfinished statement.

Fig. 2 – it is not clear why large regions of the globe that in fact have fire are not classified? For example, much of south east Asia is not classified here, despite having fire activity (see GFED map). Why is this?

Line 101 – "great match" is somewhat subjective. I suggest rephrasing or quantifying the match.

Lines 223-224: "given the disproportionate share of Tropical fires in burned area" – not clear what this means, suggest rewriting it.

Supplemental:

- Page 1 - What about areas of the world with bimodal fire seasons? This algorithm would either miss those out or assign a much shorter fire season than what happens in reality.
- Line 23 – “we sort out” should be “we sort”
- It would be useful to show Fig. S2 in the main text, perhaps as a mask to the maps shown in Fig. 2
- Lines 43-48 – why was 100ha chosen as a threshold and how robust would the results be to a different choice of threshold for fire/non-fire regions?
- In addition in Line 43, how are the authors defining “locations” – are these individual grid cells or entire regions for each climate classification?
- Line 50 and referred Figures – what exactly is the spatial unit that “places” refers to. The distributions
- Line 62-64 – I do not understand why rounding the calculated thresholds is necessary or makes them easier to use. Using the exact thresholds would presumably lead to better classification accuracy than the rounded versions, so why not keep the exact threshold values?
- Line 69 – “uncertainty” spelled incorrectly – should be “uncertainty”. In addition
- Fig. S10 – the ylabel is cut-off on panels b and d.

ANSWERS TO REVIEWER COMMENTS

All reviewers

All three reviewers have expressed concern that some regions of the planet, infrequently but notoriously affected by fires, are not classified as fire prone. This is mostly because our method identified fire prone climatic conditions from long term temperature and precipitation means, which do not adequately characterize some climates with periodic and recurring anomalies conducive to increased fire risk. A good example of this is Indonesia, which becomes fire prone in El Niño years when drought conditions make its climate more “tropical” (with dry season) than equatorial (wet year-round). Of course, all climates are susceptible to register anomalies that favor fire occurrence, but these variations are not always *expected* with a sufficiently small periodicity.

We have made a modification in the methodology to account for this deficiency, while still retaining the essence of the original method and its results. We summarize the changes as follows:

1. As suggested by reviewer #1, mean annual burned area is no longer used to identify fire prone areas. Now the burned area threshold is applied year by year, labeling for each spatial point the years with burned area above the threshold as fire-prone.
2. Similarly, we no longer use long term (20-year) monthly or annual means to identify fire prone climatic conditions, but the mean values over fire affected years.
3. The classification is made on an annual basis, so that we can compare how the fire season characteristics vary from year to year at each location. This allows us to classify certain regions that were not classified in the previous manuscript version, like Southeast Asia, most of Alaska, the Eastern United States, or Southeastern Australia, where a favorable fire setting is present mostly on certain years only. In addition, we analyze the recurrence of fire-prone conditions, so we subdivided the fire-climate classes into three subclasses, depending on how often these fire-prone conditions appear: recurrent, occasional, and infrequent.

With the new strategy, the climatic thresholds to define the main classes vary very little from what they were before, which indicates that the climatic characteristics favoring fire occurrence emerge as clearly with only data from fire affected years as with long term means, and that they are robust. What we presented in the earlier version of the manuscript, the fire prone areas identified with long term climate means, very much correspond to the combination of the recurrent and occasional subclasses. The new information comes from the “infrequent” subclass, which characterizes many previously unaccounted for well-known infrequently fire affected regions.

We thank all three reviewers for their constructive comments, which we think have improved the manuscript substantially. We hope that the following lines will respond satisfactorily to all the raised questions and comments.

The line numbering indications in these answers refer to the manuscript with tracked changes.

Reviewer #1

- *While in general, I found the analysis interesting but the use of a thresholded mean annual burned area to identify fire prone areas is very problematic because it ignores the basic fire ecology of many of these systems. Given the short time series of satellite burned area data, many areas with long fire return intervals would only likely see fire once in the entire record. That means if you average the annual burned area, many places would be characterized as having a small mean burned area but they are indeed very fire prone. Only places that burn frequently, like African savannas would appear to have a high mean annual burned area. I am concerned that*

this method failed to classify most of Alaska and Eastern Siberia, the Entire Eastern United States, Indonesia, and Southeastern Australia as fire prone. Some of these areas have infrequent but high intensity fires that heavily impact ecosystems and people. This suggests a methodology flaw that must be corrected, especially since the main conclusions about how the fire prone areas have increased shows an increase in Boreal areas such as Alaska where fire is already common and widespread. It is likely that the using a maximum annual burned area would give you a different result.

We agree with the reviewer's comment, so we have modified the methodology of our study to improve the classification in places with long fire return intervals. This has led to some variations in the results, but the main conclusions and the central idea remain unchanged. We comment here on the implemented changes:

1. Fig. 1a now represents the maximum annual burned area instead of the mean annual burned area, to better represent areas that have infrequent but high intensity fires.
 2. The classification is now made considering individual years separately, instead of using 20 year climatology means. We are not using mean annual burned area anymore, but each year's actual burned area, to separate fire years from non-fire years; and we use climatological means over either set of years, to form the distributions of fire-prone and non-fire prone data, to calculate the thresholds.
 3. We now analyze how often climatic fire-prone conditions are registered at each grid cell. We count the number of years with at least one month meeting the classification conditions, to define the number of fire-prone years (FPY), shown in Fig. 3. This strategy allows us to split the fire-prone classes into three subclasses: recurrent (r), occasional (o) and infrequent (i), giving a sense of the degree of fire risk.
 4. The new classification is now shown in Fig. 2. For the combination of recurrent and occasional classes, it very much corresponds to what we had before, which indicates that what we could identify using long term climate means was roughly areas where fire-prone climatic conditions are registered in 3 out of 10 years or more. What is new now, is that we also show regions where fire-prone climatic conditions are not so frequent (in 1 or 2 years per decade), but may very well actually translate into large fires. Most of the Alaska fire affected area, the Eastern United States, Indonesia, Siberia South Eastern Australia and other areas previously not well captured are now better reflected in the classification.
 5. The main improvement is the extra information that we get from this classification, an estimate of the recurrence of the climatic fire-prone conditions. Boreal regions are those with more infrequent fire-prone conditions, but also Central Europe or the Eastern United States show a low recurrence of these conditions. Indonesia, where fire activity is associated with El Niño events, is now classified in this new version as infrequently fire-prone, with a similar frequency to that of El Niño events.
 6. We kept the mean annual burned area representation, but it is now located in the Supplementary Material, Supplementary Fig. S2a. We also included a representation of the number of years per decade with an annual burned area greater than 100 ha per pixel, Supplementary Fig. S2b, to show the recurrence of observed fire activity worldwide.
- *Further, the use of mean annual burned area heavily biased the claim of 94.7% of fire prone areas being classified because of the overwhelming amount of burned area in regions like Sub-tropical*

Africa. Simply classifying areas as burned or unburned, rather than as a percentage of their mean annual burned area, would likely vastly reduce that classification accuracy and would highlight this deficiency.

We used the mean annual burned area to quantify accuracy because we consider more important to classify regions with high fire activity than regions with low fire activity. But we agree with the reviewer's observation in that this strategy could yield misleading results. For this reason, we have included more information about the validation of the classification in the main text of the new version: the percentage of burned area classified globally and by climate zone, and the percentage of burned/unburned area classified. By doing so, we prove that our method classifies a great percentage of the burned area in all climates (not only in the Tropics).

Additionally, we have included a figure (Supplementary Fig. S20) to better illustrate the accuracy of the classification. We now show that globally, the percentage of observed mean annual burned area associated with locations classified as fire-prone is 99%, almost all in regions with recurrent or occasional fire-prone conditions. By climate zones, those numbers are equally high, except for boreal fires, where the area captured in our classification descends to 89%. As the reviewer suggests, we also perform a validation by simply considering whether points have burned area or not, regardless of the amount. Doing so, globally we still classify 91% of points with observed burned area, and again the lowest accuracy is in boreal climates, with still as much as 82% of points classified.

Further examination of skill statistics indicates that many of the points (about 50%) that we classify wrongly as being fire prone, when no fires have been registered, are in the new category with only infrequent fire prone conditions. It is likely that the 20 year period of observations is not sufficient to cover the fire return period in those locations.

The bottom line is that a large majority of the observed burned area is well captured by our classification in the recurrent and occasional categories, globally and in all climate zones. Moreover, we classify correctly most points that have registered fires in the 20 year period considered, regardless of burned area. The false-alarms are mostly in regions that we classify as only having fire-prone conditions infrequently.

- *In the Supplement, the authors detail the method used discriminating the points with and without fire activity and they state that the thresholds were selected 'automatically' at the point that discriminates the two distributions. However, they failed to detail HOW this process works. As such, I do not believe that someone else could reproduce these results from this explanation. Additional clarification is required to ensure results are reproducible.*

We introduced five modifications in the manuscript to further clarify the threshold selection method.

1. As suggested by reviewer #3, the explanation of the classification methods was moved from the Supplementary Information to the Methods section of the main paper.
2. We rewrote some parts of the Threshold selection subsection to provide a more clear and accessible explanation on how the method works.
3. We made a scheme/cartoon (Supplementary Fig. S4) to illustrate what each statistical distribution is representing.

4. We included the analytical information on which we base the threshold selection in each statistical distribution plot (the value of the threshold and its uncertainty, the value of the area difference, the percentage of fire points meeting the threshold, the percentage of the non-fire points meeting the threshold and the percentage of burned area associated with the fire points that meet the threshold).
 5. To ensure the reproducibility we deposit the programming code used for this threshold selection process in a repository where it can be freely accessed.
- *Finally, the definition of fire season length very likely won't work for places that have a bi-modal fire season. In many places.*

Yes, we agree. Places with a bi-modal fire season could be not well identified because of the constrain of being consecutive months in the definition of the fire season. Therefore, we modified this, allowing non-consecutive months in our definition to accurately identify bi-modal fire seasons.

- *The use of A,B,C and D in the manuscript is difficult to follow and I found I needed to flip back and forth a lot to interpret parts of the manuscript. I recommend using the first two letters of the climate type (Tr for Tropical, Te for Temperate, Ar for Arid and Bo for Boreal or something similar. It should help increase readability throughout the manuscript and the supplemental information.*

We accept the recommendation. Now we use the first two letters of the climate type throughout the text, to make it more readable.

- *In the Supplementary Material, it would be more readable if the authors provided complete figure captions rather than stating "Same as Fig. S3".*

Following the reviewer's suggestion, we have written out the complete captions of the figures in the new version of the manuscript.

Reviewer #2

- *One area that needs attention is the distinction between weather and climate. Fire activity in many places depends on the day-to-day weather and is often determined by extreme conditions on a small number of days. Monthly variables may not always be good indicators of fire activity in some regions. For example, the monthly precipitation amount is not as important as frequency of precipitation. I do not require the authors to change the temporal scale of your analysis but rather acknowledge that the temporal scale may be too coarse in some regions where the bulk of the fire activity may occur over a very short time period associated with extreme conditions.*

We fully agree with the reviewer. Perhaps we haven't made clear enough that we are not searching for atmospheric indicators associated with fire risk on a particular day, but to characterize climates in which those days with fire risk, often translating into actual fires, occur with some regularity, as part of the environmental conditions in the area. The latter is what we mean by a region being fire prone.

We have now added further emphasis on that we are talking about broad spatial and temporal scales in the introduction (lines 37-40 and 53-56):

“The general conclusion emerging from these investigations is that climate related, rather than human factors, are the major controls on global fire activity **on a broad**

spatiotemporal scale, and in particular fuel availability (usually quantified by net primary production, NPP) and precipitation^{10,13}”

“The underlying hypothesis is that, **on broad spatial and decadal scales**, there is a high probability of observed fire occurrence wherever a favourable climatic fire setting exists, regardless of the nature of the ignition agent.”

It is also true that monthly values might not be a good indicator of fire risk in some places where fires only occur in extreme weather. Notwithstanding, at the global scale those locations cannot be a majority, since our method works relatively well regardless, as skill score statistics reflect. In the Tropics, with the pronounced seasonality of precipitation, high fire risk exists in the dry season without the need of any extreme conditions. In Mediterranean climates, which concentrate fire prone areas in the midlatitudes, extreme fire weather is also part of regular climate, even in the areas with milder temperatures. Mediterranean-type climates exist in the transition zone between the dry Tropics and the more humid extratropics, and experience intrinsically high variability, which often makes these areas more vulnerable to climate changes. However, in boreal regions, where fires occur in the warm but wet season, extreme fire weather conditions are likely worse captured by long term monthly climate variables, and we do see this in the poorer performance of our classification in some of those areas.

We have changed some aspects of the methodology, attempting to address the issue of climate values not identifying properly fire vulnerability in some regions. We now analyze variables in a year-to-year basis, both annual and monthly means, instead of using climatic averages of those annual and monthly values over the whole 20-year period. This change improves results mostly in areas where fire weather may only occur in years presenting some anomaly detectable at the monthly scale, like for example El Niño years in Borneo. It also reduces biases in the boreal regions, where fire months tend to be noticeable drier than average.

In this new version of the manuscript, when discussing the reliability of the results, we wrote a more detailed explanation of this issue in lines 153-160, including two examples of how the temporal resolutions can alter the results:

“In addition, as the classification is conceived from a climatic point of view, locations with fire activity associated with specific meteorological conditions that are not appreciable at the monthly temporal resolution, are probably not be well identified. For example, a week of extremely high temperatures could be almost unnoticeable in the monthly mean temperature, but not in fire activity. Similarly, months with the same total precipitation may have different fire activity if the precipitation falls concentrated in a few days or is distributed throughout the month”

- *Did you consider looking at vapor pressure deficit (VPD) that has been found to be related to fire activity? VPD drives the dead fuel drying process.*

We have calculated global maps (Fig. A1 and A2) of mean VPD in the fire season and VPD anomaly from the annual mean in the fire season, as in Fig. 1 for precipitation and temperature. Even though VPD is indeed correlated with fire activity, we do not see in these maps any added value in using VPD for identifying fire prone regions at the global scale, and we prefer to keep using common basic variables used to describe climate, such as temperature and precipitation for the task.

Figure A1. Mean monthly VPD values during the fire season.

Figure A2. Mean monthly VPD anomaly from the annual values during the fire season.

Vapor pressure deficit (VPD) is widely used in studies analyzing the meteorology of fire events, because it is well related to the ignition and daily fire spread (Seager et al 2015, Sedano et al. 2014). On a climatological scale, VPD is also a good indicator of fire activity, especially in regions with fires occurring during hot and dry summers (like the western US), because VPD is correlated with both seasonal temperature and precipitation (Holden et al. 2018). Maybe it is for this reason that on a broader global spatial scale, climatology values of VPD do not correlate better with fire activity than the common precipitation and temperature variables that we used to develop our climate-fire classification.

Holden, Z. A., Swanson, A., Luce, C. H., Jolly, W. M., Maneta, M., Oyler, J. W., ... & Affleck, D. Decreasing fire season precipitation increased recent western US forest wildfire activity. *Proceedings of the National Academy of Sciences* **115**, E8349-E8357 (2018).

Seager, R., Hooks, A., Williams, A. P., Cook, B., Nakamura, J. & Henderson, N. Climatology, variability, and trends in the US vapor pressure deficit, an important fire-related meteorological quantity. *Journal of Applied Meteorology and Climatology* **54**, 1121-1141 (2015).

Sedano, F. & Randerson, J. T. Multi-scale influence of vapor pressure deficit on fire ignition and spread in boreal forest ecosystems. *Biogeosciences* **11**, 3739-3755 (2014).

- *Inferring future change in wildland fire activity using present relationships may not work due to the likelihood that the relationships found are not stationary. This needs to be acknowledged.*

Additionally, there are lots of cells with no area burned due to a very short temporal sampling period, so this data is censored. For cells with no area burned, we do not know if the fire cycle is 30 years or 10,000 years. This limitation needs mention.

The idea of the study is to identify which locations will have the climatic conditions that are at the present associated with fire activity, but it is true that the relations can change in the future. From our analysis we cannot account for changes that may occur in climate variables at the sub monthly scale impacting fire risk, such as an increased frequency of temperature extremes or changes in precipitation distribution. We added two sentences acknowledging this limitation in lines 189-194:

“We note that we determine future fire activity from relationships of the latter with the present climate, however these relationships might not be stationary. The use of monthly variables implies that we cannot analyse changes in the frequency and intensity of extreme precipitation events in a conclusive way. The total precipitation from these intense events is increasing mainly due to changes in frequency under global warming, while the intensity changes are relatively weak³⁶.”

We also added two sentences mentioning the limitation of the burned area short temporal sampling period on lines 160-162: “The short temporal sampling period of the burned area data could also be influencing our results. Locations with long fire cycles may not be well represented in the data.”

- *Title should use wildland fire or vegetation fire instead of just fire.*

We accept the reviewer’s suggestion, and have now changed the title to “Spatial and temporal expansion of global wildland fire activity in response to climate change”.

- *Might mention the role of wildland fire smoke on human health (Johnston et al. 2012). Johnston FH, Henderson SB, Chen Y, Randerson JT, Marlier M, Defries RS, Kinney P, Bowman DM, Brauer M (2012) Estimated global mortality attributable to smoke from landscape fires. Environmental Health Perspectives 120(5), 695–7*

We thank the reviewer for the suggestion. We have now included a short sentence that mentions the role of fire smoke on global mortality, referencing the study of Johnston et al. (2012) in the introduction section (lines 29-32):

“However, fire can be a hazard to the environment, especially during extreme fire events, which have substantial economic, social and ecosystemic impacts⁷. Moreover, landscape fire smoke can be harmful to human health and is an important contributor to global mortality⁸.”

- *L57-58 these are fireless months during the fire season?*

We compared months with high fire activity at locations with fire incidence against months with low or no fire activity the rest of the year at those same locations and year-round elsewhere (cells with no fire incidence) within the same general climate class. The explanation of what we consider high fire activity and low or no fire activity can be found in the Methods section (325-355), and it is based on burned area observations. That sentence was just a very short summary of the Threshold selection Methods section to give the reader a brief explanation of how the classification was made. We now correct the brief sentence and include a further note referring the reader to the Methods section to find the whole description: “The classification is made by contrasting the probability distribution of the climatic variables at fire-affected vs. fireless months, out of the FS at locations with fire incidence plus months year-round elsewhere, within each general climate class (see Threshold selection in Methods section).”

- L174 Savanna fires are also dependent on ignitions patterns (temporal and spatial) as well as local/national policies and practices.

We have added in the new version a statement clarifying this point. In line 92-93 we included “despite savanna fires being dependent on ignition patterns and human policies and practices”.

- L104 supports rather than confirms.

We accept the suggestion, so “confirms” was changed to “supports”.

- L133 RCP8.5 – only RCP used, so should acknowledge that this is on the extreme end of the CMIP5 scenarios.

We agree. We have now included a clarification in line 167: “considering the RCP8.5 scenario (the worst-case climate change scenario of the CMIP5)”.

Reviewer #3

- *The manuscript “Spatial and temporal expansion of global fire activity in response to climate change” proposes a classification of potential fire season (PSL) based on temperature and precipitation and uses this classification to project future PSL. In general, this seems like an interesting and potentially useful approach and the results seem both interesting and relevant. However, I believe this paper needs restructuring since the most interesting aspects of the paper are currently buried in the supplemental, which describes in detail the classification of the PSL for each region. I understand Nature CC has word restrictions which may have necessitated moving many of the details to supplemental; however, the purpose of supplemental is not to supplant the main text due to word limits. Indeed, the methods section (lines 381-382) refers the reader to the supplemental for all details of threshold selection, and Tables 1 and 2 lack context in the main paper with a better description in the main text.*

We understand and share the reviewer’s concern. In fact, our first idea was to explain all the details of the threshold selection process in the main paper, but at the same time we wanted to present a brief main text, so we had decided to move that part to the Supplementary Information. In this new version of the manuscript, per the reviewer’s suggestion, the threshold selection method is explained in the main text, specifically on the Methods section, lines 325-394. Unfortunately, the statistical distributions on which we base the threshold selection method cannot be shown in the main paper due to the editorial limitation on the number of figures. Thus, we think that the specific discussions on each distribution fit better accompanying the figures to which they refer, in the Supplementary Information. The main message of such discussions is nevertheless included in the main text, when the different classes are introduced. We hope to present a more readable and interesting main text with this reorganization of the article.

- *The manuscript considers precipitation and temperature thresholds to calculate potential fire season. Did the authors also consider atmospheric humidity or surface wind speeds in their analysis, which are well-known to correlate with fire danger? (see Canadian Fire Weather Index system for example).*

Surface wind speed is an essential variable for fire spread, but from a climatic point of view it is not that relevant. During windy days fires can spread quickly, so fire activity can be enhanced, but the fire season of a region is hardly determined by the average wind conditions (see Figures A5 and A6 below). Similarly, atmospheric humidity modifies the fine fuel moisture, favoring fire spread, but on a broader spatial and temporal scale this variable loses significance (Figures A3

and A4 below). In fact, the Canadian Fire Weather Index splits the fuel moisture calculations into three codes with different temporal terms (2/3, 12 and 52 days of drying rate), and the variables used to calculate each code are different, with only temperature and precipitation determining the long-term part of the fuel moisture (Van Wagner 1974). In summary, although surface wind and atmospheric humidity are very important variables in *fire meteorology*, they are not so important for *fire climatology*, which is the focus of our study. Our approach of not using the wind and humidity variables to characterize fire climate on such a broad temporal and spatial scale is usually shared by other studies (e.g. Aldersley et al. 2011, Andela et al. 2017, Krawchuck et al. 2009, Moritz et al. 2012 or Van Der Werf et al. 2008).

Figure A3. Mean monthly 2m relative humidity (RH) anomaly from the annual values during the fire season.

Figure A4. Mean monthly RH values during the fire season.

Figure A5. Mean monthly 10m wind speed anomaly from the annual values during the fire season.

Figure A6. Mean monthly 10m wind speed values during the fire season.

Aldersley, A., Murray, S. J. & Cornell, S. E. Global and regional analysis of climate and human drivers of wildfire. *Sci. Total Environ.* **409**, 3472-3481 (2011).

Andela, N. et al. A human-driven decline in global burned area. *Science* **356**, 1356-1362 (2017).

Krawchuk, M. A., Moritz, M. A., Parisien, M. A., Van Dorn, J. & Hayhoe, K. Global pyrogeography: the current and future distribution of wildfire. *PloS ONE* **4**, e5102 (2009).

Moritz, M. A. et al. Climate change and disruptions to global fire activity. *Ecosphere* **3**, 1-22 (2012).

Van Der Werf, G. R., Randerson, J. T., Giglio, L., Gobron, N. & Dolman, A. J. Climate controls on the variability of fires in the tropics and subtropics. *Global Biogeochem. Cycles* **22**, GB3028 (2008).

Van Wagner, C. E. *Structure of the Canadian forest fire weather index*. (Ontario: Environment Canada, Forestry Service, 1974).

- *It is not clear what the spatial units are for the modeling of probability density functions for climatic variables (and assessment of thresholds). It seems to be done at each of the four main*

regions (A, B, C, and D), which is rather broad spatially. However, the supplemental includes figures S5-S8 that show the associations for smaller spatial units. How robust are the results to choice of these spatial units given that fire activity and climate are both heterogenous even within a single region (eg. each A, B, C or D category). Did the authors consider using smaller spatial units, while understanding the spatial units should be large enough to encompass a well-defined fire season.

Yes, the probability density functions were computed for each of the four main climatic regions. Some Supplementary Figures are focusing on specific regions just to understand some issues, but not to calculate any thresholds. Reducing the spatial scale allowed us to better adjust the global classification results to the observed burned area data. However, the main goal of the study is to use broad spatial scales to bring out more clearly the climatic characteristics of fire prone regions, providing a global perspective. The main result is that indeed fire risk and occurrence follow common climatic patterns across global scales, that is to say that the characteristics of climate in fire prone regions are shared within each broad global climatic zone and tend not to be specific of a particular region.

If we defined thresholds for each spatial point, we would have a classification totally adjusted to the burned area data, but many thresholds would not respond to natural climatic mechanisms but to human ones. Moreover, with such classification, it would be very difficult to assess the circumstances of different regions, especially those with no observed burned area. On the contrary, if we analyze the entire planet as a single category, many fire climate characteristics would remain difficult to precise. Therefore, we have had to find a balance between these two options.

We tested some modifications consisting in reducing the spatial scale, unsuccessfully trying to get better results in the boreal area, where we obtain a poorer performance. For example, splitting the Boreal climate spatially between American Boreal and Eurasian Boreal would lead to the result shown in Figure A7 (applying the same method to both subdivisions). As we see, results are not so different from those in the original classification using data from both areas combined, so we dismissed this modification. In turn, this result is a confirmation that there are indeed common fire climate characteristics across global scales within the same general climate zone. Another option for reducing the spatial scale would be to use all of the Köppen-Geiger climate divisions, but the criteria defining the 2nd and 3rd letter of the different climates would overlap with our potential fire season definitions, because some are based on temperature and precipitation seasonality.

Figure A7. Boreal climate splitting. Classified fire-prone boreal regions applying the threshold selection method to the American boreal region and Eurasian boreal region separately.

- *The climate projections presented here assume that monthly precipitation changes will dictate future PFSs within each category. However, much of the literature now points towards changes in both intensity and frequency of future precipitation events (eg. Myhre et al. 2019). How would changes in precipitation frequency change the results reported here? The authors should address this, at least in the discussion. Myhre, G., Alterskjær, K., Stjern, C.W. et al. Frequency of extreme precipitation increases extensively with event rareness under global warming. *Sci Rep* 9, 16063 (2019). <https://doi.org/10.1038/s41598-019-52277-4>*

We agree with the reviewer’s comment, which in fact is related to an issue also pointed out by reviewer #2, who remarks that we should acknowledge that inferring future change in wildland fire activity using present relationships may not work due to the likelihood that these relationships are not stationary. The projected future change in intensity and frequency of extreme precipitation events is an example of a reason why the relationship between mean monthly precipitation and fire risk might possibly be non-stationary, as it can depend on precipitation distribution as well.

The idea of the study is to identify which locations will have in the future the climatic conditions that are associated with fire activity at the present, but it is true that the relations can change in time. We included the following paragraph in lines 189-194, acknowledging this issue and referencing the study of Myhre et al. (2019):

“We note that we determine future fire activity from relationships of the latter with the present climate, however these relationships might not be stationary. The use of monthly variables implies that we cannot analyse changes in the frequency and intensity of extreme precipitation events in a conclusive way. The total precipitation from these intense events is increasing mainly due to changes in frequency under global warming, while the intensity changes are relatively weak³⁶. ”

Myhre, G., Alterskjær, K., Stjern, C.W. et al. Frequency of extreme precipitation increases extensively with event rareness under global warming. *Sci Rep* **9**, 16063 (2019).
<https://doi.org/10.1038/s41598-019-52277-4>

- *Lines 18: What impacts do wildfires have? It wouldn't hurt to elaborate here.*

See response to comment on Line 24 below

- *Lines 20-21: "biosphere by biomass burning" seems quite vague – what changes to the biosphere are occurring?*

We included in this new manuscript version two examples of biosphere changes produced by fires (lines 24-25): deforestation and modification of the vegetation structure.

- *Line 24: "Fire can be a hazard..." reads better as "However, fire can be a hazard..."*

We followed the previous three reviewer's recommendations and changed the paragraph accordingly as follows (lines 21-32):

"Global fire patterns are determined by climate and fuel availability, along with the existence of ignition agents and human factors¹. At the same time, fires also modify climate through the emission of aerosols and greenhouse gases, and the biosphere by biomass burning, which can lead to deforestation or other modifications of the vegetation structure^{2,3}. Indeed, wildfires produce important impacts on habitats and societies worldwide⁴. As a natural process, fire plays a role in some ecosystems, such as being a regulator of biomass in savanna biomes⁵. It is also used as a management tool in pastoral and agricultural areas with regular ignitions from humans⁶. However, fire can be a hazard to the environment, especially during extreme fire events, which have substantial economic, social and ecosystemic impacts⁷. Moreover, landscape fire smoke can be harmful to human health and is an important contributor to global mortality⁸."

- *Lines 27-28 and lines 31-34: I would say these statements are not entirely true since the dominant controls on wildfire are very regionally dependent; in many regions of Asia and Africa fire depends predominantly on human activity since many of the ignitions are human-caused. For example see: Forkel, M., Dorigo, W., Lasslop, G., Chuvieco, E., Hantson, S., Heil, A., ... & Harrison, S. P. (2019). Recent global and regional trends in burned area and their compensating environmental controls. *Environmental Research Communications*, 1(5), 051005.*

We modified that part of the manuscript (lines 37-42), remarking that we are analyzing fire activity on a broad spatiotemporal scale and included a sentence mentioning that the controls on wildfire activity can vary at a regional scale and human activity can be a major control in some areas (Forkel et al. 2019):

"The general conclusion emerging from these investigations is that climate related, rather than human factors, are the major controls on global fire activity on a broad spatiotemporal scale, and in particular fuel availability (usually quantified by net primary production, NPP) and precipitation^{10,13}. However, on a smaller, regional scale, the drivers of wildfire activity are more varied, and human activity can be the major factor in some areas¹⁶."

Forkel, M., Dorigo, W., Lasslop, G., Chuvieco, E., Hantson, S., Heil, A., ... & Harrison, S. P. Recent global and regional trends in burned area and their compensating environmental controls. *Environmental Research Communications* **1**, 051005 (2019).

- *Lines 45-47: Fire occurrence is often ignition- or fuel-limited despite favourable climatic conditions; I would suggest the authors emphasize that they are talking about over a sufficiently long time period (ie. Decades).*

We accepted the suggestion, so we indicate that we are talking about broad spatial and temporal scales (line 54).

- *Fig. 1b. I would suggest not using a diverging colormap for this panel since the quantities do not diverge around zero. Also, with respect to this plot, why not show precipitation anomalies instead of percentage difference?*

We changed the colormap of Fig. 1b to a non-diverging colormap. The reason why we use precipitation percentages instead of precipitation anomalies is that, due to the high differences in precipitation values between world regions, the anomalies can lead to misleading conclusions. For example, a precipitation anomaly of -50 mm in the tropical rainforest is not so intense as a precipitation anomaly of -50 mm in Siberia. We think that representing the precipitation percentage from the annual value avoids these issues.

- *Lines 68-72 – why is it important to distinguish between FS and PFS here and what are the implications? This text seems like an unfinished statement.*

We decided to include that sentence because we thought that it could be confusing to the reader to talk about potential fire season and fire season alongside in the whole text. The potential fire season is the one that we obtain from our analysis of the climate data, and the fire season is the one that is obtained from the burned area data. However, since this distinction was already explained above in the text, we removed this sentence in the new manuscript version.

- *Fig. 2 – it is not clear why large regions of the globe that in fact have fire are not classified? For example, much of south east Asia is not classified here, despite having fire activity (see GFED map). Why is this?*

Some of the unclassified regions were discussed in the Supplementary Information, like Southeastern Australia or Southeast Asia. In particular, the latter is only fire prone during EL Niño years, a circumstance that was not well detected using climatic means over a 20 year period. Many of these unclassified regions are now classified after the modifications introduced in the classification following another reviewer's suggestions, that have substantially improved the results. We now use year to year data (monthly and annual means) instead of 20-year averages to apply the thresholds, and can therefore determine how often (in terms of yearly frequency) a certain region is fire prone. Southeast Asia is now classified in the new classification with a recurrence that is similar to the periodicity of El Niño events.

- *Line 101 – “great match” is somewhat subjective. I suggest rephrasing or quantifying the match.*

We agree with the comment, so we included an extensive quantification of the match in the Supplementary Fig. S20b. We added in the main text a sentence to direct the reader to the Supplementary Fig. S20b.

- *Lines 223-224: “given the disproportionate share of Tropical fires in burned area” – not clear what this means, suggest rewriting it.*

We mean that most of the burned area worldwide lies in the Tropics. We rewrote the sentence to “given the vast contribution of Tropical fires to the burned area at the global scale”. We hope it is now easier to understand.

- *Page 1 - What about areas of the world with bimodal fire seasons? This algorithm would either miss those out or assign a much shorter fire season than what happens in reality.*

Yes, probably some regions with bimodal fire seasons were not being well identifying. We modified the fire season definition in order to avoid this, so we remove the constrain of being consecutive months in the definition.

- *Line 23 – “we sort out” should be “we sort”*

We accept the suggested modification.

- *It would be useful to show Fig. S2 in the main text, perhaps as a mask to the maps shown in Fig. 2.*

We tried to include the Köppen-Geiger classes in Fig. 2, but due to the classification modifications of this new version, with more fire-climate classes, we did not find a suitable way to include that information in the figure.

- *Lines 43-48 – why was 100ha chosen as a threshold and how robust would the results be to a different choice of threshold for fire/non-fire regions?*

We assessed the robustness of the method by applying the same procedure but using different burned area thresholds and analyzing how different the resulting classifications are. As we see in Fig. A8a, the percentage of points that change from a fire-prone class to unclassified (or vice versa) is less than 2.5% for all tested thresholds up to 2500 ha. Including all changes from one fire-prone class to another, the variations are always in less than 10% of the points.

In Fig. A8b and A8c we show the final classification when using the minimum possible threshold of 0 ha, and for 2500 ha. As we see, the regions classified are quite similar, with very smooth differences between both classifications. This means that the method is quite robust because the main conclusions would remain unchanged when using different thresholds.

We note that now the selected threshold of 100ha is applied year by year, and not to the mean annual burned area, addressing a comment of reviewer #1.

- *In addition in Line 43, how are the authors defining “locations” – are these individual grid cells or entire regions for each climate classification?*

When we use the word “locations” we refer to individual grid cells. We rewrote that paragraph to better explain the methodology, also avoiding the use of the word locations when referring to grid cells.

- *Line 50 and referred Figures – what exactly is the spatial unit that “places” refers to. The distributions.*

In that line, places refer to those individual grid cells with a cropland land cover percentage of more than 90%. We now clarify this in the text.

- *Line 62-64 – I do not understand why rounding the calculated thresholds is necessary or makes them easier to use. Using the exact thresholds would presumably lead to better classification accuracy than the rounded versions, so why not keep the exact threshold values?*

We know that the sensitivity of the classification results to smooth modifications of the thresholds (like rounding them) is quite low. For example, analysing the uncertainties of the thresholds, the classification differences quantified as the percentage of points that change from a fire-prone class to unclassified between the most restrictive thresholds (adding the uncertainty to the exact thresholds values for T_a , T_{max} , T_m and P_a and subtracting the uncertainty to the exact thresholds values for P_{min} and P_m) and the less restrictive thresholds (subtracting the uncertainty to the exact thresholds values for T_a , T_{max} , T_m and P_a and adding the uncertainty to the exact thresholds values for P_{min} and P_m) is only 1.5%. This allows us to select a threshold value inside the uncertainty range being confident that the results will be very similar to those using the exact threshold values. The objective is to select threshold values without many decimals, e.g. for the monthly temperature threshold of the Arid climate we obtain a threshold of $19.67 \pm 0.25^\circ\text{C}$ and we select 19.5°C . Usually this type of climate classifications, like the Köppen-Geiger classification are based on rounded values.

- *Line 69 – “uncertainty” spelled incorrectly – should be “uncertainty”.*

Yes, it was a typing error. That part of the supplementary material is no longer included in the manuscript, but we checked that all the “uncertainty” words are well spelled.

- *In addition, Fig. S10 – the ylabel is cut-off on panels b and d.*

We checked that all the panels of the statistical distributions are fully visible.

a
Figure A8. Burned area threshold robustness analysis. **a**, Comparison between the results using 100 ha of burned area threshold and other possible thresholds. In black, the percentage of points classified (unclassified) with the tested threshold and unclassified (classified) with 100 ha threshold. In grey, the percentage of points with any classification difference between the tested threshold and the 1000 ha threshold. **b**, Classified fire-prone regions using 0 ha as the burned area threshold. **c**, Classified fire-prone regions using 2500 ha as the burned area threshold.

REVIEWER COMMENTS

Reviewer #2 (Remarks to the Author):

I want to thank the authors for their detailed and thoughtful response to the review comments. I have just a few more comments.

The Delta method is used for future temperature and precipitation but this approach allows for changes in precipitation amounts but not for precipitation frequency that could be important for fire in some regions as you mention in the manuscript. This assumption and possible implications needs to be stated.

The paper states there is no ignition limitation at coarse spatial and temporal resolutions (L127-130). However, see Keeley et al. 2021 but you might consider this to be spatially localized.

L240-242 This is a significant result and you may want to expand on this in terms of feedbacks as there are large areas of peatlands that have significant legacy carbon banks. These large stores of legacy carbon that could be emitted to the atmosphere through combustion.

Editorial

I find lines 67-69 confusing

L314 Which?

Keeley, J.E., Guzman-Morales, J., Gershunov, A., Syphard, A. D., Cayan, D., Pierce, D.W., Flannigan, M. and Brown, T.J. 2021. Ignitions explain more than temperature or precipitation in driving Santa Ana wind fires. Science Advances 21 Jul 2021. Vol. 7, no. 30, eabh2262. DOI: 10.1126/sciadv.abh2262

Reviewer #3 (Remarks to the Author):

The manuscript is much improved after a substantial revision. In particular, the change in methodology for identifying fire prone years and the resulting fire season is an important and interesting addition. Moving the description of threshold selection into the methods also greatly improves the presentation. However, there are a few issues I would still like to see resolved before I can recommend publication. These are listed below. I refer to line numbering in the marked-up revision.

Line 17 – suggest changing “lengthenings” to “lengthening”.

Lines 25-26 – the sentence “Indeed, wildfires produce important impacts on habitats and societies worldwide.” seems redundant given the rest of the paragraph. It can be removed without loss of information.

Lines 55-56 – I am not sure it is accurate to qualify this sentence with “regardless of the nature of the ignition agent.” as ignition agents will not be spatially random themselves. There may be areas of preference for lightning caused fires due to underlying climatology factors and for human caused fires due to recurrent human activity on the landscape and population density considerations.

Line 68 – I cannot follow the meaning of the added text “out of the FS at locations with fire incidence plus months year-round elsewhere,”. This should be rewritten.

Lines 82-83 – how exactly are the categories “recurrent (r), occasional (o) and infrequent (i)” defined? I see this is defined in the methods so perhaps add “(see methods)” or similar here.

Line 119: I still do not like the term “great match” because it is quite subjective and the supplementary supporting material (Fig. S20b) does not necessarily support this because of the number of false positives - for example. I might be better to quote the area that is correctly classified with $BA > 0$ or $BA = 0$ and leave it to the reader to decide if it is great.

Fig 3 – panel A. No sure how the numbers on the colorbar match up with a difference in fire season length (months) since the numbers are shown at the breaks. For example is yellow a difference of +1 months or +2 months? The same question applies also to Fig. 4 panel a.

Lines 365-367 – “Distributions with a big area difference indicate a distinct behaviour of the values associated with fire activity with respect to the values that are not.” I do not fully follow this sentence – are you saying there is greater interannual variability for these regions?

Lines 374-380 – this information should be included in the corresponding figure caption and not here.

ANSWERS TO REVIEWER COMMENTS

All reviewers

We thank all reviewers for these new constructive comments, which we think they have further improved the manuscript. We hope that the following lines will respond satisfactorily to all the raised questions and comments.

The line numbering indications in these answers refer to the manuscript with tracked changes.

Reviewer #2

- The Delta method is used for future temperature and precipitation but this approach allows for changes in precipitation amounts but not for precipitation frequency that could be important for fire in some regions as you mention in the manuscript. This assumption and possible implications needs to be stated.

We agree with the reviewer's suggestion, so we changed lines 189-197 to mention this issue and to point out what we consider to be its main implication for our study:

"Our approach does not contemplate possible future changes in precipitation frequency if they are not noticeable in monthly precipitation amounts. Areas with rising incidence of extreme precipitation events due to global warming³⁷ may experience an increase in monthly precipitation but a decrease in rainy days, which may lead us to consider the conditions there less favorable for fire activity than they actually will be."

- The paper states there is no ignition limitation at coarse spatial and temporal resolutions (L127-130). However, see Keeley et al. 2021 but you might consider this to be spatially localized.

Keeley, J.E., Guzman-Morales, J., Gershunov, A., Syphard, A. D., Cayan, D., Pierce, D.W., Flannigan, M. and Brown, T.J. 2021. Ignitions explain more than temperature or precipitation in driving Santa Ana wind fires. *Science Advances* 21 Jul 2021. Vol. 7, no. 30, eabh2262. DOI: 10.1126/sciadv.abh2262

We agree that for certain events and even for certain locations, ignitions can be a more important factor than atmospheric conditions. We have modified the referred lines to make this issue more precise (L125-132):

"Although ignitions may be driving fires to a greater extent than temperature or precipitation at specific locations or events (Keeley et al., 2021), they do not seem to limit fire activity at coarse spatial and temporal resolutions, implying that, where fuels are sufficient and atmospheric conditions are conducive to combustion, the potential for ignition exists, either by lightning or human causes^{13,20}."

- L240-242 This is a significant result and you may want to expand on this in terms of feedbacks as there are large areas of peatlands that have significant legacy carbon banks. These large stores of legacy carbon that could be emitted to the atmosphere through combustion.

Thank you for your comment. We have included two sentences explaining this possible feedback (L239-243):

"This possible increase in fire activity in boreal areas may result in significant peatland combustion and a release of the large quantities of soil carbon that they store into the atmosphere (Nelson et al., 2021). These greenhouse gas emissions will create a positive feedback loop, leading to a further increase in temperature, which in turn will enhance boreal wildfire incidence and more peatland burning".

Nelson, K., Thompson, D., Hopkinson, C., Petrone, R., & Chasmer, L. Peatland-fire interactions: A review of wildland fire feedbacks and interactions in Canadian boreal peatlands. *Science of the Total Environment* **769**, 145212 (2021).

- [Editorial] I find lines 67-69 confusing

These lines are intended as a brief preview of the methodology used to obtain the thresholds, which is explained in detail in the Methods section. We have modified those lines to a simpler statement and added a sentence in brackets referring the reader to the Methods section (L63-69).

"The classification is made by contrasting the probability distribution of the climatic variables at data points associated with high fire activity vs. points with low fire activity within the main Köppen-Geiger categories (see Threshold Selection in Methods section for a detailed explanation)."

- [Editorial] L314 Which?

Thank you for the correction. It was a typing error that has been fixed in this new version of the manuscript (L314).

Reviewer #3

- Line 17 – suggest changing “lengthenings” to “lengthening”.

We accept the reviewer's suggestion, so “lengthenings” has been changed to “lengthening” (L16).

- Lines 25-26 – the sentence “Indeed, wildfires produce important impacts on habitats and societies worldwide.” seems redundant given the rest of the paragraph. It can be removed without loss of information.

We agree with the comment, so we have removed the referred sentence (L23-24).

- Lines 55-56 – I am not sure it is accurate to qualify this sentence with “regardless of the nature of the ignition agent.” as ignition agents will not be spatially random themselves. There may be areas of preference for lightning caused fires due to underlying climatology factors and for human caused fires due to recurrent human activity on the landscape and population density considerations.

We agree that ignition agents are not spatially random themselves. That part of the sentence was aimed at emphasizing that ignitions do not seem to limit fire activity at coarse spatial and temporal resolutions. However, to avoid a possible confusion, we removed that part of the sentence in this new version of the manuscript (L52-53).

- Line 68 – I cannot follow the meaning of the added text “out of the FS at locations with fire incidence plus months year-round elsewhere,”. This should be rewritten.

These lines are intended as a brief preview of the methodology used to obtain the thresholds, which is explained in detail in the Methods section. We have modified those lines to a simpler statement and added a sentence in brackets referring the reader to the Methods section (L63-69).

"The classification is made by contrasting the probability distribution of the climatic variables at data points associated with high fire activity vs. points with low fire activity within the main Köppen-Geiger categories (see Threshold Selection in Methods section for a detailed explanation)."

- Lines 82-83 – how exactly are the categories “recurrent (r), occasional (o) and infrequent (i)” defined? I see this is defined in the methods so perhaps add “(see methods)” or similar here.

We accept the suggestion, so we added “(see Methods)” in that line (L80).

- Line 119: I still do not like the term “great match” because it is quite subjective and the supplementary supporting material (Fig. S20b) does not necessarily support this because of the number of false positives - for example. I might be better to quote the area that is correctly classified with $BA > 0$ or $BA = 0$ and leave it to the reader to decide if it is great.

We agree with your suggestion. That sentence is probably an unscientific and vague statement. We have now rewritten the text in a more rigorous way, as proposed in your comment (L112-120):

"The correspondence between these two maps is quantified in Supplementary Fig. 20b, with more than 70% of the land area well classified as either fire prone ($BA > 0ha$) or fireless ($BA = 0ha$). This reveals a two-way relation between fires and climate: fires take place under specific climatic conditions, and most places with these climatic conditions are indeed fire prone, which supports our earlier hypothesis."

- Fig 3 – panel A. No sure how the numbers on the colorbar match up with a difference in fire season length (months) since the numbers are shown at the breaks. For example is yellow a difference of +1 months or +2 months? The same question applies also to Fig. 4 panel a.

We used colorbars with the numbers shown at the breaks for continuous quantities, and with the numbers shown at half color bins for discrete quantities. Although the variable in Fig. 3 (difference in fire season length) has units of months, it is continuous because it comes from an average of several years and different model outputs. Therefore, it takes values in between whole integer numbers, and it is better plotted with a colorbar with labels at the breaks. What may lead to confusion is that colorbars of panels B and C do not show numbers at the breaks, even though they also represent continuous quantities. We corrected this issue, so now the colorbars of all the figures are coherent.

- Lines 365-367 – “Distributions with a big area difference indicate a distinct behaviour of the values associated with fire activity with respect to the values that are not.” I do not fully follow this sentence – are you saying there is greater interannual variability for these regions?

Not exactly. We agree that the sentence is confusing. What we mean is that when distributions associated with and without fire incidence for any given variable are very different from each other, they indicate that fires occur in places with clearly distinct values of such variable. And that the inverse is also true, places that do not register those values do

not have fires, or have them to a much lesser extent. The climatic variables or indicators that we examine to bring out the climatic drivers of fires are related to precipitation and temperature, but can be of different types (annual, monthly, fire season averages) and are explained in the Methods section with detail.

We have now rewritten the sentence to make the point clearer (L365-369):

“Distributions with a big area difference evidence that the considered variable is a good indicator of fire incidence, since the values associated with fire activity are clearly distinct from the values that are not.”

- Lines 374-380 – this information should be included in the corresponding figure caption and not here.

We accept the reviewer's suggestion, so we moved that text to the corresponding figure caption.

REVIEWERS' COMMENTS

Reviewer #2 (Remarks to the Author):

Thanks for your thorough response to the review comments. I have only 2 minor comments.

L239-243

"This possible increase in fire activity in boreal areas may result in significant peatland combustion and a release of the large quantities of soil carbon that they store into the atmosphere (Nelson et al., 2021). These greenhouse gas emissions will create a positive feedback loop, leading to a further increase in temperature, which in turn will enhance boreal wildfire incidence and more peatland burning".

for the last sentence suggest gas emissions may create as opposed to will create.

Lastly, there is a recent published that may be relevant to this paper - see below.

Jain, P., Castellanos-Acuna, D., Coogan, S.C.P. Abatzoglou, J.T., and Flannigan, M.D. 2021. Observed increases in extreme fire weather driven by atmospheric humidity and temperature. Nature Climate Change. <https://doi.org/10.1038/s41558-021-01224-1>

Reviewer #3 (Remarks to the Author):

The authors have satisfactorily addressed my concerns and I believe the paper is suitable for publication.

One small issue remains in the modified sentence, lines 367-269 (line numbers for track change version) - I believe there may be a missing word before "evidence". Also do the authors mean to say "good indicator of fire incidence" rather than "if"?

ANSWERS TO REVIEWER COMMENTS

All reviewers

We thank all reviewers for these new comments. We hope that the following lines will respond satisfactorily to all the raised comments. Some extra corrections were made in the manuscript text to comply with some editorial requests. The line numbering indications in these answers refer to the manuscript with tracked changes.

Reviewer #2

Thanks for your thorough response to the review comments. I have only 2 minor comments.

- L239-243 "This possible increase in fire activity in boreal areas may result in significant peatland combustion and a release of the large quantities of soil carbon that they store into the atmosphere (Nelson et al., 2021). These greenhouse gas emissions will create a positive feedback loop, leading to a further increase in temperature, which in turn will enhance boreal wildfire incidence and more peatland burning". For the last sentence suggest gas emissions may create as opposed to will create.

We accepted the suggestion, so we changed "will create" for "may create" in line 203.

- Lastly, there is a recent published that may be relevant to this paper - see below. Jain, P., Castellanos-Acuna, D., Coogan, S.C.P. Abatzoglou, J.T., and Flannigan, M.D. 2021. Observed increases in extreme fire weather driven by atmospheric humidity and temperature. *Nature Climate Change*. <https://doi.org/10.1038/s41558-021-01224-1>
Thank you for the comment. We included this article as a reference in line 216 where we mention the increase of the extreme fire weather in the Western US during the las decades, that is one of the main results of the article.

Reviewer #3

The authors have satisfactorily addressed my concerns and I believe the paper is suitable for publication.

- One small issue remains in the modified sentence, lines 367-269 (line numbers for track change version) - I believe their may be a missing word before "evidence". Also do the authors mean to say "good indicator of fire incidence" rather than "if"?

Thank you for the comment, it is actually a typing error. We changed the word "if" by "of" and we added the article "a". The sentence would read as follows (lines 327-329):

"Distributions with a big area difference evidence that the considered variable is a good indicator of fire incidence, since the values associated with fire activity are clearly distinct from the values that are not."

We think that the sentence is now well written. There is no missing word before "evidence", because in this case it acts as a verb and not as a noun. Because of the way the sentence was written in the previous version, it may be confusing and it may appear that the word "evidence" acts as a noun.